# Green Assessment of Phenolic Acid Composition and Antioxidant Capacity of Advanced Potato Mutant Lines through UPLC-qTOF-MS/MS Quantification

**DOI:** 10.3390/foods12132616

**Published:** 2023-07-06

**Authors:** Clara Gomez-Urios, Hristo Kalaydzhiev, Jesus Blesa, Maria Jose Esteve, Emiliya Nacheva, Dida Iserliyska, Nasya Tomlekova

**Affiliations:** 1Nutrition and Food Science, Faculty of Pharmacy, University of Valencia, Avenida Vicent Andrés Estellés, s/n, 46100 Burjassot, Spain; clara.urios@uv.es (C.G.-U.); jesus.blesa@uv.es (J.B.); maria.jose.esteve@uv.es (M.J.E.); 2Department of Analytical Chemistry and Physical Chemistry, University of Food Technologies—Plovdiv, 26 Maritsa Blvd., 4002 Plovdiv, Bulgaria; hristo.kalaydzhiev@yahoo.com; 3Maritsa Vegetable Crops Research Institute, Agricultural Academy—Sofia, 32 Brezovsko shosse St, 4003 Plovdiv, Bulgaria; emnach@abv.bg; 4Institute of Food Preservation and Quality—Plovdiv, Agricultural Academy—Sofia, 154 Vasil Aprilov Blvd., 4003 Plovdiv, Bulgaria; dida_isser@yahoo.com

**Keywords:** mutant potato, tuber, ultrasound-assisted extraction, green chemistry, antioxidant capacity

## Abstract

Potatoes are one of the most consumed crops worldwide. They contain a high amount of bioactive compounds such as phenolic compounds and vitamins with important antioxidant activities, which makes this crop of high biological value for human health. The goal of this research was to biochemically evaluate polyphenol levels and antioxidant capacities in parent and control genotypes compared to advanced mutant potato lines in the M_1_V_8_ generation. This will reveal the genetic changes that result from induced mutagenesis. The quantified compounds and the evaluated antioxidant activity boost the health benefits of consuming the improved mutant potatoes. In the present study, the phenolic composition and antioxidant activity of eighteen mutant and initial potato genotypes were analyzed by UPLC-qTOF-MS/MS and the ORAC method, respectively. In each of the hybrid combinations, mutant lines with an improved phenolic compound profile were observed. Representative samples from the third hybrid combination had notable increases in phenolic compound concentrations, as well as the presence of metabolites not found in the parental lines. With one exception, the remaining nine mutants showed significantly higher antioxidant capacities. The results will be used in future potato breeding programs, with participation of the valuable mutant lines containing new phenolic substances not present in the initial genotypes.

## 1. Introduction

Phenolic acids, as one of the main groups classified under phenols of plant origin, are considered to be a dietary component of major importance for human health through their tremendous antioxidant activity. Antioxidants are an inhibitor of the oxidation process, even at relatively small concentrations, and thus have diverse physiological roles in the body. A variety of free radical scavenging antioxidants is found in dietary sources like fruits, vegetables, and tea [1]. Epidemiologic evidence indicates that a diet rich in antioxidant fruits and vegetables significantly reduces the risk of many oxidative-stress-related diseases viz. cancers, diabetes, and cardiovascular diseases [2]. Vegetables are easily accessible, abundant in quantity, and economical sources of phenolic acids and thus will have a high impact on human health worldwide. Awareness regarding this will enhance the overall well-being of the consumer. A range of health benefits have been attributed to phenolic acids and an increase in the consumption of vegetables can be used to derive these benefits [3].

The contributors to the health benefits of potatoes are mainly linked to the presence of triterpenoid glycosides, antioxidant flavonoids, and carotenoids. As good sources of phenolic compounds present mainly in the potato skin and flesh, potato tubers contain several bioactive compounds, including polyphenolics (e.g., chlorogenic acid, methylbelliferones, and the flavonoids apigenin, rutin, and kaempferol 3-O-rutinoside), which have demonstrated activity as antimicrobial agents [2] and agents against cancer [4,5,6], heart disease [7], hypertension [8], diabetes [9], Parkinson’s disease [10], and Alzheimer’s disease [11,12].

According to the reaction process, the techniques for determining antioxidant capacity can be roughly divided into two groups: techniques based on hydrogen atom transfer (HAT) and techniques based on electron transfer (ET) [13]. The oxygen radical antioxidant capacity (ORAC) assay is the most popularly used HAT method. The net protection area under the duration-recorded fluorescence decline curve is used in the ORAC assay to quantify the antioxidant capacity when an antioxidant is present in the sample.

Potatoes may contain more genetic diversity than any other crop and this may reflect the ability of potatoes to grow in remarkably divergent environments [14]. This genetic diversity is a valuable resource for further improvement of the tuber’s nutritional content [15]. Mutation breeding is a potentially powerful tool for potato improvement [16]. Chemical mutagenesis is an important way to cause genetic variation [17]. The described mutants are obtained by induced mutagenesis and subsequent mutation breeding, combined with standard analytical methods. Induced mutagenesis is a method that resembles the natural mutation process that occurs in plants. The primary distinction is that it occurs more frequently than spontaneous mutations in nature. These mutant lines can be used as a source of new material in traditional breeding programs. The genetic diversity enhancement of potato (*Solanum tuberosum* L.) by sexual hybridization and induced mutagenesis using the chemical mutagen ethyl methane sulfonate (EMS), having an impact on the breeding programs in Bulgaria, have been applied by some authors [18,19]. The methods of induced mutagenesis and sexual hybridization were used to enrich the genetic diversity of potato mutant lines that differ significantly in morphological and economic characteristics both between each other and in comparison with the parental components and the untreated controls [19].

The benefits of consuming phenolic compounds for human health are huge, but the process to obtain them should be eco-friendly. Lately, innovative techniques for extracting phenolic compounds as well as for determining antioxidant capacity have been reported [20]. Thus, the use of ultrasound-assisted extraction (UAE) has been demonstrated to be a good alternative to conventional extraction due to it offering time and energy savings, higher yields, and secure, high-quality extracts [21], making the extraction process sustainable.

In the current study, a total of 10 advanced mutant potato lines were previously subjected to induced mutagenesis and thereafter to mutation breeding, and selected by phenotype and genotype by quality- and quantity-altered mutant traits. The selected mutants with increased productivity are M-I-8, M-I-17, M-III-8, M-III-9, M-III-30, M-III-48, M-III-50, M-IV-14, M-IV-17, and M-VII-7. They differed from the parental lines (PC 428, PC 490, PC 538, PC 707, and PC 757) and each other. All mutant and parental lines, as well as all control samples, were analyzed for biochemical components with health implications. The compounds that were investigated were chlorogenic acid (CGA), narirutin (NR), naringin (NIN), hesperidin (HSP), ferulic acid (FA), naringenin (NAR), hesperetin (HET), trans-cinnamic acid (CA), kaempferol (KMP), apigenin (API), quercetin (QUE), and rutin (RT). The goal of this research was to evaluate polyphenol levels biochemically and the antioxidant capacity in parent and control genotypes compared to advanced mutant potato lines in the M_1_V_8_ generation. This reveals the genetic changes that result from induced mutagenesis.

## 2. Materials and Methods

### 2.1. Plant Material

The study included 10 promising mutant potato lines developed at the Institute of Vegetable Crops “Maritza”-Plovdiv, 5 parental components (cultivars and lines), and 3 controls (not mutant lines). The previously investigated parameters (length of the vegetation period, plant height, number of stems per plant, standard size, nonstandard size, total number of tubers per plant, standard, nonstandard, and total tuber weight per plant, and average weight of a standard size tuber with increased productivity) have been reported and were used for selecting only beneficial mutant traits [19]. The methodological path is shown in Figure 1.

After a metabolic profiling procedure, mutants with improved plant/tuber phenotypes and enhanced total phenolic concentrations of the so-called first (I), third (III), fourth (IV), and seventh (VII) hybrid combinations were selected. The hybrid combinations, parental genotypes, and origins of the mutant lines and controls are presented in Table 1. The mutant lines were developed by treating F_1_ hybrid potato seeds with the chemical mutagen 0.3% ethylmethane sulfonate (EMS) and the controls were untreated F_1_ hybrid seeds from the same hybrid combinations [19]. All treated seeds were sown and M_1_ plants were grown. The resulting tubers from each M_1_ plant were harvested separately and recorded as a separate mutant. Further multiplication of these mutants continued vegetatively. Untreated seeds (control) were sown under the same conditions, control plants were grown, and the tubers of these plants were collected separately and registered as controls. After preliminary selection for significant morphological and economic traits, 10 promising mutant potato lines (M_1_V_8_) were selected from four hybrid combinations to continue the mutation breeding process. For each parental and mutant (M_1_V_8_) line, samples were collected from tubers in triplicate.

Due to the length of the mutant development process and independent circumstances of the environment during multiplication, the PC 692 parental line was not available for this study, and neither was the control sample from the first hybrid combination. Only the best selected controls were selected for maintenance. Since PC 428 and PC 490 parental lines were used in two hybrid combinations, they are displayed in Table 2, Table 3 and Table 4 to make it easier to evaluate the outcomes. The summarized table (Appendix A) for all samples tested is available in the Appendix A.

### 2.2. Chemicals and Reagents

The following analytical standards were used to identify and quantify polyphenols: chlorogenic acid ≥ 95.0% (CGA), narirutin ≥ 98% (NR), naringin ≥ 95.0% (NIN), hesperidin ≥ 97.0% (HSP), ferulic acid ≥ 99.0% (FA), naringenin ≥ 95% (NAR), hesperetin ≥ 98.0% (HET), trans-cinnamic acid ≥ 98.0% (CA), and kaempferol ≥ 97% (KMP). These were purchased from Sigma-Aldrich (Steinheim, Germany). Apigenin 96.86% (API), quercetin dihydrate 97.02% (QUE), and rutin trihydrate 97.67% (RT) were obtained from HWI Analytik GmbH (Ruelzheim, Germany).

Analysis grade ethanol, HPLC grade methanol (MeOH), analysis grade hydrochloric acid, and HPLC grade acetonitrile were purchased from J.T. Baker Chemical Co. (Deventer, The Netherlands). Analysis grade formic acid was purchased from PanReac AppliChem (Darmstadt, Germany). Ultrapure water (18.2 MΩ cm) was used for preparing the mobile phase, and other aqueous solutions were obtained using a Milli-Q water purification system (Millipore, Molshein, France). C18 (200/6 mg·mL^−1^) cartridges (55 μm, 70 Å) for SPE were purchased from Phenomenex (Steinheim, Germany). Nylon membrane filters (0.22 μm) were obtained from Millex, Millipore (Tullagreen, Carrigtwohill, Ireland). Trolox (±)-6-hydroxy-2,5,7,8-tetramethyl chromate-2-carboxylic acid), 2,2′-azobis(2-amidinopropane) dihydrochloride (AAPH), and sodium fluorescein were purchased from Sigma-Aldrich (Steinheim, Germany).

### 2.3. Standard Preparation

The guidelines were created utilizing the method by Anticona et al. [22]. The standards (400 µg/mL) were prepared in MeOH 0.2% formic acid. Working solutions were prepared by diluting each standard in 75% MeOH (*v*/*v*) and 0.6 M HCl up to 50 µg/mL. According to each standard, a concentration in the range of 0 and 50 g/mL was prepared for the calibration curves. Every standard was run through a 0.22 mm nylon filter.

### 2.4. Ultrasound-Assisted Extraction (UAE)

An ultrasonic processor Q500 (Qsonica, Melville, NY, USA) was employed for ultrasound-assisted extraction (UAE), with the working frequency set at 20 kHz. The extraction process used 100 W, a 20% (*v*/*v*) duty cycle, and a maximum temperature of 40 °C. A small glass containing 10 g of lyophilized potatoes was filled with 100 mL of aqueous ethanol (50%, *v*/*v*), which served as the solvent. With a 30 s ultrasound pulse and 59 s of recovery, samples were ultrasonically treated for 10 min. Each extract was centrifuged after treatment, and the supernatant was then gathered in a volumetric flask that was shielded from light. Until they were required for analysis, all the extracts were kept at 4 °C.

### 2.5. Solid Phase Extraction (SPE)

The clean-up procedure used to separate the polyphenols from the potato extracts was based on the SPE reported by Anticona et al. [22] using C18 (200 mg) cartridges. A 5 mL of each extract of 0.1% formic acid was diluted with 20 mL of Milli-Q water quickly.

After the cartridges had been preconditioned by being filled twice with 3 mL of methanol and twice with 3 mL of MilliQ water, samples were placed into them. Solvents and samples were left in the cartridges for two minutes prior to loading. After the samples were loaded, the cartridges were cleaned with 5 mL of 0.1% formic acid in Milli-Q water. The polyphenol fractions were eluted using 3 mL of 0.1% formic acid in MeOH. All of the methanolic fractions were redissolved in 1 mL of HCl (0.6 mol/L) after being lightly nitrogen dried.

### 2.6. Polyphenol Quantifications–UPLC-qTOF-MS/MS

For phenolic compound separations, confirmatory studies were carried out using a TripleTOFTM 5600 LC/MS/MS system (SCIEX) utilizing an ACQUITY UPLC BEH C18 column (2.1 50 mm I.D., 1.7 m; Waters). The following elution binary gradient was utilized with a flow rate of 0.4 mL.min^−1^: 0–5 min, isocratic 70% solvent A (water/formic acid 99.9/0.1 (*v*/*v*)) and 30% solvent B (methanol/formic acid, 99.9/0.1 *(v*/*v*)); 5–12 min, linear from 70 to 5% A; 12–18 min, isocratic 5% A; 18–18.5 min, linear from 5 to 70% A; and 18.5–25 min, isocratic 70% A. The injection had a 5 µL volume. The following instrument settings were used to capture data in negative mode over a mass/charge range of 100–950 *m*/*z*: ion source gases 1 and 2, 50 psi; curtain gas 1, 25 psi; 450 °C; and ion spray voltage, −4500 V. An external calibrant delivery system (CDS), which injects a calibration solution before the entry of the sample, was used to carry out automated calibration.

Polyphenol standards with concentrations ranging from 1 to 5 g/mL were employed for external calibration. Sample dilution was applied when the sample concentration was higher than the calibration curve. The information-dependent acquisition (IDA) approach was employed by the mass spectrometer (MS) with the survey scan type (TOF-MS) and the dependent scan type (production) utilizing a collision energy of 30 V. PeakViewTM 2.2 software, was used to qualitatively analyze the data, and MultiquantTM–3.0.3 software was utilized to quantify the phenolic chemicals found. Results were given in units of µg/mL of extract. A table of validation parameters is available in theas Appendix A.

### 2.7. Antioxidant Activity: ORAC

An oxygen radical absorbance capacity assay (ORAC), with a few modifications by Zulueta et al. [23], was carried out as reported by Ou et al. [24]. It was performed with fluorescence filters and an excitation wavelength of 485 nm and an emission wavelength of 535 nm (Wallac 1420 VICTOR2 multilabel counter apparatus, Perkin-Elmer, Waltham, MA, USA). Measurements were made using 96-well multi-well plates with a white flattened bottom (Thermo Fisher^®^, Waltham, MA, USA). A stock solution of fluorescein was prepared after dissolving 44 mg in 100 mL of phosphate buffer (75 mM, pH 7.0). Extemporaneously, 0.167 mL of the fluorescein stock solution was diluted in 25 mL of phosphate buffer to create the working solution (78 mM). Each day, 600 mg of 2,2′-azobis(2-amidinopropane) dihydrochloride (AAPH) and 10 mL of phosphate buffer were used to create the AAPH radical (221 mM).

Samples must be diluted properly due to the high sensitivity of the *ORAC* assay; in this instance, a 1:250 dilution was sufficient. Amounts of 50 μL of fluorescein (78 nM), 50 μL of the sample, and 50 μL of phosphate buffer blank or standard (20 μM Trolox) were then added to each well, followed by 25 μL of radical AAPH (221 mM). The relative fluorescence intensity was measured until it was less than 5% of the initial value of fluorescence, which was performed directly after the addition of the reagents. The measurements were taken in cycles of 5 min. *ORAC* values were stated in terms of *mM* Trolox equivalent (TE)/g dry weight (*DW*) of samples. The *ORAC* was determined using Equation (1) as follows:(1)$$ORAC\;\left(mM\;TE/\;g\;DW\right)=\left(\frac{AUC_{S}-AUC_{B}}{AUC_{TRX}-AUC_{B}}\right)*k*\alpha *h,$$
where *k* is the dilution factor, *α* is the molar concentration of Trolox, and *h* is the ratio of the sample’s volume to mass. *AUC_S_* stands for the area under the curve of the sample, *AUC_B_* for the area under the curve of the blank, and *AUC_TRX_* for the area under the curve of Trolox.

### 2.8. Statistical Analysis

Measurements were carried out in triplicate and were reported as means ± and standard deviation. SPSS 26.0.0^®^ was used for the statistical analysis (SPSS Inc., Chicago, IL, USA). An analysis of variance (ANOVA) was used to assess mean differences with a significance threshold of 0.05. Additionally, a Tukey’s test (*p* < 0.05) was used to determine between which level differences occurred for parameters with more than two levels and significant differences.

## 3. Results

### 3.1. Polyphenol Quantifications–UPLC-qTOF-MS/MS

A novel method for chromatographic separations, ultra-high performance liquid chromatography-quadrupole time-of-flight mass spectrometry (UHPLC/Q-TOF-MS), was used effectively for quick, high-resolution separations with adequate sensitivity. TOF-MS is useful for determining the fragmentation patterns of substances and elucidating their structures [25]. In the tables presented below, the results for p-coumaric acid, caffeic acid, and catechin were removed due to the complete absence of these compounds in our samples, which corresponds with findings of other authors [26,27,28].

The results in Table 2 show that ferulic acid was not present in one of the mutant lines (M-I-17), but hesperetin was detected only in this line (0.0180 ± 0.0005 µg/mL). The other mutant line, M-I-8, is the only one of the lines presented in Table 2 in which quercetin was detected (0.1015 ± 0.0005 µg/mL). Trans-cinnamic acid was present only in the parental sample PC 428 (5.1700 ± 0.0020 µg/mL). Rutin, narirutin, and quercetin were present only in the mutant plants. Apigenin, chlorogenic acid, hesperidin, naringenin, and kaempferol were detected in all samples. It is noteworthy that the level of chlorogenic acid in the parental plants was higher compared to the mutants but remains the phenolic compound with the highest concentration in the samples examined. The concentration of kaempferol in the mutant line M-I-8 (0.0025 ± 0.0002) was statistically higher compared to parental lines and the other mutant line M-I-17. The results for naringenin in the mutants did not differ from those in the parental lines and were in the range of 0.0070 to 0.0080 µg/mL.

The control (K-III-2) of the third hybrid combination (Table 3) was characterized by higher concentrations of phenolic compounds compared to the parental line (PC 490), in which rutin, naringin, and hesperetin were not even detectable. Neither quercetin, narirutin, or trans-cinnamic acid were detected in the parental line nor in the control plant. Ferulic acid, apigenin, chlorogenic acid, hesperidin, naringenin, and kaempferol were present in all samples. Ferulic acid (0.0895 ± 0.0005 µg/mL) and naringenin (0.0210 ± 0.0002 µg/mL) were highest in M-III-48, while a higher concentration of quercetin (0.4165 ± 0.0010 µg/mL) was detected in the mutant line M-III-30. From the results obtained (Table 3), it is clear that the chlorogenic acid concentrations in M-III-50 (9.2200 ± 0.0021 µg/mL) and M-III-48 (7.5400 ± 0.0021 µg/mL) were higher than all other samples, including the parental line and control. Narirutin was present in only two mutant lines, M-III-8 and M-III-48, with a higher concentration in the latter (0.0405 ± 0.0004 µg/mL). Trans-cinnamic acid was the most abundant phenolic compound in M-III-50 (1.4905 ± 0.0007 µg/mL), followed by M-III-30 (0.5090 ± 0.0014 µg/mL). It should be noted that naringenin was present in higher concentrations in the M-III-48 and M-III-30 mutants. The level of kaempferol in the control sample was the highest (0.0130 ± 0.0006 µg/mL), followed by the mutant line M-III-30 (0.0080 ± 0.0002 µg/mL).

In the fourth hybrid combination (Table 4), no amounts of naringin, hesperetin, or narirutin were detected in the parental plants, the control, and the two mutant lines. It is noteworthy that quercetin was present in the highest amount in the control sample K-IV-3 (0.0470 ± 0.0008 µg/mL), followed by mutant lines, which had statistically smaller concentrationa of this compound. Rutin was present only in the mutant lines and one of the parents, PC 707. The opposite is true for hesperidin, which was not found in the mutant lines but only in the parent plants and the control. Trans-cinnamic acid was present in the highest concentration in one of the parents, PC 428 (5.1700 ± 0.0020 µg/mL). It was not detected or was in low concentrations in the rest of studied genotypes.

In the seventh hybrid combination tested (Table 5), naringin, hesperetin, quercetin, and narirutin were not detected in any of the samples. Rutin was present only in the control sample (K-VII-4) and the mutant line (M-VII-7), with lower concentration in the mutants. Apigenin in all samples ranged from 0.0385 to 0.0440 µg/mL. Chlorogenic acid ranged from 1.8965 to 2.7125 µg/mL, with the highest concentration in the control. The hesperidin levels were lower in the control and mutant lines compared to one of the parents (0.0635 ± 0.0010 µg/mL). Trans-cinnamic acid was detected only in the M-VII-7 mutant line (0.0140 ± 0.0004 µg/mL). The naringenin levels in the mutant line were two times lower compared to the parents, and this compound was not detected in the control. Kaempferol had the highest concentration in the control sample −0.0030 ± 0.0001 µg/mL.

The selected promising genotypes with combinations of a large number of phenolic substances are two mutants. The first is M-III-48, with increased concentrations of ferulic acid, naringin (non-detected in other except of a control), hesperetin (detected only in three genotypes), chlorogenic acid (a total of four genotypes had very high concentrations compared to the rest), narirutin (present in a total of four genotypes), and naringenin (the highest concentration among all studied).

The second valuable mutant is M-III-30, with ferulic acid, rutin, apigenin (with the highest level compared to the rest), quercetin (the highest levels were detected in eight genotypes among 18 studied), hesperidin, naringenin (second ranked by levels of this substance), and kaempferol (second after M-III-48).

### 3.2. Antioxidant Capacity

The oxygen radical absorbance capacity (ORAC) test is one of the most prominent antioxidant screening procedures. The data in Table 6 clearly show the higher antioxidant activity of the mutant lines compared to the parents. The exception is the M-III-30 mutant, which has an antioxidant activity lower than that of the corresponding parental line.

With ORAC values ranging from 3.32 to 3.84 mM TE/g, the mutant lines M-I-8, M-III-9, M-III-48, and M-III-50 stood out as potential mutant lines with significant antioxidant activities. M-IV-17 and M-VII-7 mutants were also reported with ORAC values of 3.03 ± 0.41 and 3.08 ± 0.50 mM TE/g, respectively.

The results will be used in the future potato breeding programs with participation of the valuable mutant lines containing new phenolic substances not present in the initial genotypes.

## 4. Discussion

### 4.1. Phenolic Acid Composition

The mutagenic effect of EMS can be inferred from the obtained phenolic compound concentration data obtained by UPLC-qTOF-MS/MS. As previously reported by Mahpara et al. [29], the treatment of plant material with EMS was better than treatments with other chemical mutagens, providing more variants of mutants. Although, according to the same authors, the degree of success depends heavily on the amount of the chemical and the type of plant used in the experiment, chemical mutagens can successfully change a plant’s genetic composition. Potatoes have a rich biodiversity both as a plant and food. The broad nutritional and bioactive diversity of potatoes supports an assumption that tubers will have a wide range of nutrients and bioactive compounds [12]. Numerous potato populations studied had different levels of carotenoids, total phenolics, chlorogenic acid, caffeic acid, rutin, and kaempferol [30,31]. From the data in Table 2, Table 3, Table 4 and Table 5, as well as in the summary Appendix A, ferulic acid was not detected in one of the parent plants (PC 538) and one of the mutants (M-I-17). The amount of ferulic acid in parental line PC 428 was the highest (0.1145 ± 0.0004 µg/mL), but the concentration of this acid in the mutant lines was significantly lower (M-I-8–0.0210 ± 0.0002 µg/mL; M-I-17–N/A; M-IV-14–0.0095 ± 0.0004 µg/mL and M-IV-17–0.0220 ± 0.0003 µg/mL) for hybrid combinations where PC 428 was used as a parent. In the third hybrid combination, the control sample contained a higher amount of ferulic acid (K-III-2–0.0565 ± 0.0004 µg/mL) than the parental line (PC 490–0.0205 ± 0.0003 µg/mL), but this may be due to the possibly high ferulic acid concentration of the other parental line, which was not available for analysis. In the third and seventh combinations, the mutant lines M-III-9, M-III-30, M-III-48, M-III-50, and M-VII-7 were characterized by higher concentrations of ferulic acid, at 0.0655 ± 0.0005, 0.0800 ± 0.0005, 0.0895 ± 0.0005, 0.0560 ± 0.0005, and 0.0585 ± 0.0001 µg/mL, respectively, compared to the parental line.

Rutin was absent in all of the parent plants except for PC 707, where it was found at a concentration of 0.0110 ± 0.0001 µg/mL. Two of the control samples (K-III-2 and K-VII-4) from the third and seventh hybrid combinations and all mutant lines contained rutin in different concentrations. The rutin concentration in the control sample K-III-2 (0.2020 ± 0.0008 µg/mL) and the mutant line M-III-30 (0.0795 ± 0.0007 µg/mL) was prominent among all the results. Although the mutation did not lead to higher amounts of rutin in the mutant lines than in the control obtained by a simple hybridization process, its presence in the mutant lines is an indication of the mutagenic action of EMS. As reported by various authors, rutin has many biological and pharmacological properties, including antioxidant, antidiabetic, cardiovascular, and anti-inflammatory activities [32,33,34].

Naringin was not detected in either parent but was present in one control sample (K-III-2–0.0040 ± 0.0003 µg/mL) and one of the mutant lines (M-III-48–0.0015 ± 0.0002 µg/mL) of the same hybrid combination, with a lower concentration in the mutant. The situation with hesperetin was almost the same. It was found in the same control and mutant line in higher amounts of 0.0260 ± 0.0004 µg/mL and 0.0155 ± 0.0007 µg/mL, respectively, compared to naringin. Hesperetin was found in one more mutant: M-I-17 (0.0180 ± 0.0005 µg/mL). We can assume that after crossing the two parental components, in which the substances in question are undetectable in very low levels, in the controls and in the mutants, they can be established as they accumulate.

Apigenin was detected in all samples, and the mutant line M-III-30 possessed the highest concentration at 0.3200 ± 0.0013 µg/mL. The control samples of the third and fourth hybrid combinations had higher apigenin concentrations than the parental lines. Numerous research works conducted over the years have shown that apigenin has a wide range of intriguing pharmacological actions and the potential for use in nutraceuticals. For instance, its antioxidant effects are widely known, and it can also be used as a therapeutic agent to treat conditions including inflammation, autoimmune and neurological conditions, and even a few different types of cancer [35,36].

The most prevalent phenolic component in potatoes is chlorogenic acid, which accounts for around 80% of all phenolic acids [12,37]. Among all samples, the parental line PC 428 had the highest concentration of 12.5750 ± 0.0028 µg/mL. The chlorogenic acid concentration of the remaining parent lines ranged from 1.8965 to 3.7970 µg/mL. The control sample of the third hybrid combination K-III-2 had the highest chlorogenic acid concentration (6.8700 ± 0.0007 µg/mL) among the other controls, the concentration of which was 2.1180 ± 0.0008 and 2.7125 ± 0.0012 µg/mL for K-IV-3 and K-VII-4, respectively. More interesting, however, is the chlorogenic acid concentration of the mutant lines. Table 2 and Appendix A clearly show the concentration of this metabolite in M-III-48 (7.5400 ± 0.0021 µg/mL) and M-III-50 (9.2200 ± 0.0021 µg/mL), which is higher than that in the parental line PC 490 and the control sample of the same hybrid combination (K-III-2). It is possible that the other parental line PC 692, which is not available for analysis, contains more chlorogenic acid like PC 428, but this is not necessary. Since we used a chemical mutagen, this change in chlorogenic acid concentration might be due to a mutation. The remaining mutants had chlorogenic acid concentrations that ranged from 1.6415 to 2.8435 µg/mL, with the exception of M-IV-14, having the lowest level at 0.7860 ± 0.0008 µg/mL. Here, “chlorogenic acid” specifically refers to 5-O-caffeoylquinic acid, which has been thoroughly researched [38]. The primary food sources of CGAs are potato tubers, sweet potato leaves, artichokes, eggplant, sunflower seed kernels, and green coffee beans, which are considered to have the greatest chlorogenic acid concentration among any plants [39,40,41]. Chlorogenic acid and its metabolites were detectable in the blood circulation of both humans and animals in a distinct experimental setting. After regular consumption of CGA, the systemic circulation of three main chlorogenic acid metabolites—ferulic, isoferulic, and caffeic acid—was investigated [38,42,43].

Interesting results are obtained for quercetin. This compound was not detected in parental lines. One of the control samples (K-IV-3) contained 0.0470 ± 0.0008 µg/mL of quercetin. The mutant line M-III-30 had the highest quercetin concentration of 0.4165 ± 0.0010 µg/mL, followed by M-I-8 at 0.1015 ± 0.0005 µg/mL. The quercetin concentration of the remaining mutant lines ranged from 0.0170 to 0.0305 µg/mL. Only three of the mutant lines did not contain quercetin (M-I-17, M-III-9, and M-VII-7). The plant kingdom synthesizes flavonoids, which are naturally occurring substances that are typically found in foods consumed by most people [44]. Quercetin is a plant pigment that is mostly present in citrus fruits, grapes, cherries, berries, onions, green beans, and broccoli. It is a potent antioxidant flavonoid and more particularly a flavonol. It is a flexible antioxidant with the ability to protect against tissue damage induced by different medication toxicities [45,46]. As reported by Aydin et al. [47], the bioactive substance known as quercetin, which is present in numerous plants, stimulates particular transcription factors and sirtuin proteins to promote the synthesis of antioxidant enzymes that fight ROS, boost stress tolerance, and initiate the DNA repair process.

Narirutin is formed after the glycosylation of naringenin with the disaccharide rutinose [48]. Narirutin supplementation is effective in the treatment of obesity, high blood pressure, type 2 diabetes, and metabolic syndrome, according to a number of experimental studies [49]. More research on narirutin along with other flavonoids shows that they are insulin stimulating and antihypertensive in animal models [50]. Bioactive substances such as vitamin C, folate, hesperidin, narirutin, and naringin play a crucial part in preserving the reliability of immunological barriers and assisting immune cell activity [47]. Among our samples, this flavonol was detected only in four of the mutant lines from the first and third hybrid combinations (M-I-8, M-I-17, M-III-8, and M-III-48), with the greatest concentration in the last one (0.0405 ± 0.0004 µg/mL). The primary sources of narirutin are citrus fruits, water mint, cherries, oregano, tomatoes, chocolate, etc. [48]. Its absence in the parental lines and controls of two hybrid combinations can both be inferred from the process of mutagenesis.

Hesperidin, which is a glycosylated form of hesperetin, was not found in one of the parental lines (PC 538) and two of the mutant lines (M-IV-14 and M-IV-17). Flavanones have received less attention than flavonols and isoflavones, despite the fact that their dietary consumption can be substantial and they show an intriguing biological activity [46]. As reported by Pietta et al. [46], little is known regarding the absorption or kinetic behavior of the flavanones naringenin and hesperetin, as well as their glycosylated forms naringin, hesperidin, and narirutin. In more recent studies, it has been stated that hesperidin has antidiabetic, cardioprotective, and hypolipidemic potential [51,52,53,54]. Of all the mutant lines tested, a higher hesperidin concentration compared to the parental lines was reported in M-III-8 (0.1100 ± 0.0013 µg/mL) and M-III-48 (0.1025 ± 0.0007 µg/mL). The control sample of the same hybrid combination (K-III-2) contained the same amount of the metabolite as one of the mutant lines (0.1025 µg/mL). In the fourth hybrid combination, the greatest amount of hesperidin was observed in the parental lines (PC 707: 0.0120 ± 0.0003 µg/mL and PC 428: 0.0275 ± 0.0004 µg/mL), a smaller amount was found in the control sample (K-IV-3: 0.0025 ± 0.0003 µg/mL), and an absence was found in the mutant lines. In the latter hybrid combination, the amounts of hesperidin in the mutants were less than that in the parental line.

Trans-cinnamic acid was detected in some of the samples. Interestingly, cinnamic acid was not detected in any of the control samples and in three of the parental lines. It is known that their derivative is ferulic acid and both metabolites—cinnamic and ferulic acids—were absent in two samples (PC 538 and M-I-17) (Appendix A). A higher concentration of trans-cinnamic acid in mutants was observed for M-III-50, at 1.4905 ± 0.0007 µg/mL. Inference of mutational changes can be made for the last hybrid combination, where cinnamic acid was detected only in the mutant line M-VII-7. In their review article, Ruwizhi and Aderibigbe [55] have very well systematized the effects of cinnamic acid on organisms by in vitro techniques. These include anticancer, antibacterial, and antioxidant activities, and neurological and antidiabetic activities have also been reported for some derivatives.

Naringenin is an anti-inflammatory, anticarcinogenic, and antitumor flavonoid aglycone of naringin. It has also been shown to have anti-estrogenic properties and may influence several oxidative processes linked to chronic degenerative conditions [56]. As it is shown in summary Appendix A, naringenin is not found in only one sample: control K-VII-4. Higher concentrations were observed in samples M-III-48 (0.0210 ± 0.0002 µg/mL) and M-III-30 (0.0180 ± 0.0004 µg/mL), which exceeded the naringenin quantity in the control and parental lines.

Kaempferol was present in all samples in small quantities. In parental lines, it was in a concentration range of 0.0010–0.0015 µg/mL as in most of the other samples. The highest amount of kaempferol was recorded in control K-III-2 (0.0130 ± 0.0006 µg/mL), followed by mutant line M-III-30 (0.0080 ± 0.0002 µg/mL).

### 4.2. Antioxidant Activity

This test is founded on the hypothesis that the compound 2,20-azobis(2-amidinopropan) dihydrochloride generates reactive oxygen radicals (ROSs), initiated by thermal decomposition, with the ability to oxidize fluorescein. This substance fluoresces intensively, in contrast to the extremely weak fluorescence of its oxidation product. As a result of the ROSs oxidizing the fluorescein, the solution’s light fades over time. Fluorescein will not oxidize until an antioxidant is used, because when an antioxidant is introduced to the test, it interacts with the ROSs [57]. Trolox in a concentration of 20 μM was used as a standard and phosphate buffer was used as a blank probe. The antioxidant activity of the samples was determined by the highly sensitive ORAC method [23,24], but the discussion in this study will be limited to the relationship between antioxidant activity and the presence of the phenolic compounds detected. As can be seen in Table 6, the mutant lines are characterized by a higher antioxidant activity compared to the parental lines, the only exception being the mutant line M-III-30, which recorded the lowest antioxidant activity of all samples. The highest antioxidant capacity was reported in the mutant lines of the third hybrid combination, M-III-9 (3.81 ± 0.30 mM TE/g), M-III-48 (3.84 ± 0.31 mM TE/g), and M-III-50 (3.41 ± 0.38 mM TE/g), which were statistically indistinguishable from each other. There is a mutant line present across every hybrid combination that has the highest antioxidant capacity compared to parental lines. For the first hybrid combination, this is M-I-8 (3.32 ± 0.13 mM TE/g), whose antioxidant capacity is higher than the other mutant (M-I-17–2.61 ± 0.40 mM TE/g). A significant difference was also observed in the two mutant lines of the fourth hybrid combination, M-IV-14 and M-IV-17, in favor of the latter. In the seventh hybrid combination, the results for antioxidant capacity were very similar, with a statistical difference observed only in the parental line PC 538 (smallest) and the mutant line M-VII-7 (highest).

The first thing noticed was the presence of quercetin in the mutant lines, which was absent in all the parental plants. As mentioned earlier in the paper, quercetin has a strong antioxidant activity, which could have an effect on the antioxidant capacity of the mutant lines. Rutin is a strong antioxidant and is present in all mutant lines. It is noteworthy that narirutin, which is absent in parental lines and controls, is present in the mutant line with the highest antioxidant capacity, M-III-48 (3.84 ± 0.31 mM TE/g), which demonstrates the impact of mutagenesis. In the same sample, the highest amount of naringenin was detected.

Data on the antioxidant capacity of the samples varied to some extent, and some samples with higher phenolic concentrations showed a low antioxidant capacity. It has been shown that numerous vitamins, carotenoids, and phenylpropanoids are the most prevalent antioxidants in potato tubers [58]. Potatoes contain carotenoids, which affect the flavor and aroma as well as plant resistance to photo and oxidative stress [59,60]. The probable concentration of these substances in the tested extracts may influence their antioxidant capacity and interfere with the discussion when attempting to relate the amounts of phenolic compounds and antioxidant capacity.

## 5. Conclusions

The mutant lines obtained have comparable amounts of phenolic compounds with the parental lines and controls, but the true effect of mutagenesis is seen in the mutant lines containing metabolites not present in the parental lines. An outstanding mutant was M-III-48, in which the greatest increases in levels of phenolic compounds were reported, followed by M-III-50, which also possessed a balanced profile of phenolic acids.

These two mutant lines also exhibited a high antioxidant capacity, along with M-I-8, M-III-9, M-IV-17, and M-VII-7. Another promising mutant was M-III-30, which possessed a remarkable profile of phenolic compounds but had the lowest antioxidant capacity of all the mutant lines.

Further investigation is needed to determine the remaining metabolites possessing antioxidant activity to explain the controversial data on the phenolic compound concentration and antioxidant capacity of the advanced mutant potato lines.

## Figures and Tables

**Figure 1 foods-12-02616-f001:**
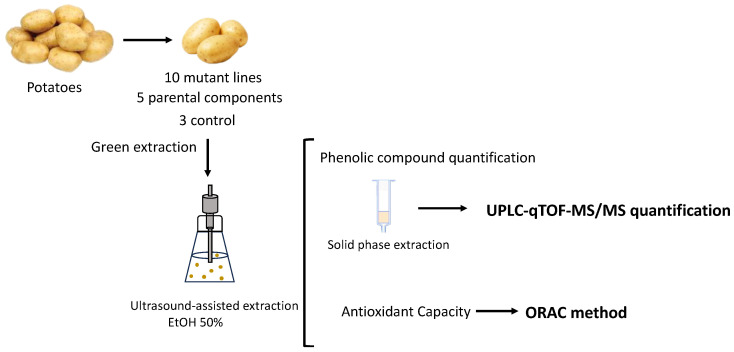
Methodological path.

**Table 1 foods-12-02616-t001:** Parental genotypes, mutant lines, and controls: origin, groups, and number of mutants.

Mutant Group	Hybrid Combination	Origin	M_1_V_8_ Mutant Lines	Controls
M-I	PC 428 × PC 490	“Nadezhda” × I 75.127 N	M-I-8, M-I-17	N/A
M-III	PC 692 (N/A) × PC 490	“Orlik” × I 75.127 N	M-III-8, M-III-9, M-III-30, M-III-48, M-III-50	K-III-2
M-IV	PC 707 × PC 428	“Olza” × “Nadezhda”	M-IV-14, M-IV-17	K-IV-3
M-VII	PC 757 × PC 538	E 402 × “Karlena”	M-VII-7	K-VII-4

N/A: not available.

**Table 2 foods-12-02616-t002:** Quantity of phenolic compounds in parent and advanced potato mutant lines (M_1_V_8_) from the first hybrid combination.

Concentration, (µg/mL)				
Phenolic Compound	PC 428	PC 490	M-I-8	M-I-17
Ferulic acid	0.1145 ± 0.0004 ^a^	0.0205 ± 0.0003 ^b^	0.0210 ± 0.0002 ^b^	N/D
Rutin	N/D	N/D	0.0075 ± 0.0005 ^b^	0.0240 ± 0.0004 ^a^
Naringin	N/D	N/D	N/D	N/D
Apigenin	0.0430 ± 0.0004 ^c^	0.0445 ± 0.0009 ^b^	0.0985 ± 0.0005 ^a^	0.0430 ± 0.0003 ^c^
Hesperetin	N/D	N/D	N/D	0.0180 ± 0.0005
Chlorogenic acid	12.5750 ± 0.0028 ^a^	3.7970 ± 0.0009 ^b^	2.2790 ± 0.0010 ^c^	1.7350 ± 0.0006 ^d^
Quercetin	N/D	N/D	0.1015 ± 0.0005	N/D
Narirutin	N/D	N/D	0.0080 ± 0.0004 ^a^	0.0085 ± 0.0003 ^a^
Hesperidin	0.0275 ± 0.0004 ^d^	0.0710 ± 0.0010 ^a^	0.0325 ± 0.0003 ^c^	0.0590 ± 0.0006 ^b^
Trans-cinnamic acid	5.1700 ± 0.0020	N/D	N/D	N/D
Naringenin	0.0075 ± 0.0003 ^a^	0.0070 ± 0.0002 ^ab^	0.0080 ± 0.0002 ^a^	0.0080 ± 0.0002 ^a^
Kaempferol	0.0015 ± 0.0001 ^b^	0.0015 ± 0.0001 ^b^	0.0025 ± 0.0002 ^a^	0.0010 ± 0.0002 ^c^

^a–d^ Means in a row with the same lowercase letter do not differ significantly (*p* ≥ 0.05). N/D—not detected.

**Table 3 foods-12-02616-t003:** Quantity of phenolic compounds in parent, control, and advanced potato mutant lines (M_1_V_8_) from the third hybrid combination.

Concentration, (µg/mL)							
Phenolic Compound	PC 490	K-III-2	M-III-8	M-III-9	M-III-30	M-III-48	M-III-50
Ferulic acid	0.0205 ± 0.0003 ^f^	0.0565 ± 0.0004 ^d^	0.0220 ± 0.0005 ^e^	0.0655 ± 0.0005 ^c^	0.0800 ± 0.0005 ^b^	0.0895 ± 0.0005 ^a^	0.0560 ± 0.0005 ^d^
Rutin	N/D	0.2020 ± 0.0008 ^a^	0.0295 ± 0.0001 ^c^	0.0025 ± 0.0003 ^f^	0.0795 ± 0.0007 ^b^	0.0110 ± 0.0004 ^e^	0.0150 ± 0.0005 ^d^
Naringin	N/D	0.0040 ± 0.0003 ^a^	N/D	N/D	N/D	0.0015 ± 0.0002 ^b^	N/D
Apigenin	0.0445 ± 0.0009 ^bc^	0.0465 ± 0.0007 ^b^	0.0400 ± 0.0006 ^d^	0.0395 ± 0.0004 ^d^	0.3200 ± 0.0013 ^a^	0.0455 ± 0.0004 ^bc^	0.0430 ± 0.0004 ^c^
Hesperetin	N/D	0.0260 ± 0.0004 ^a^	N/D	N/D	N/D	0.0155 ± 0.0007 ^b^	N/D
Chlorogenic acid	3.7970 ± 0.0009 ^d^	6.8700 ± 0.0007 ^c^	1.9030 ± 0.0006 ^f^	1.6415 ± 0.0014 ^g^	2.8435 ± 0.0008 ^e^	7.5400 ± 0.0021 ^b^	9.2200 ± 0.0021 ^a^
Quercetin	N/D	N/D	0.0170 ± 0.0002 ^c^	N/D	0.4165 ± 0.0010 ^a^	0.0170 ± 0.0004 ^c^	0.0185 ± 0.0003 ^b^
Narirutin	N/D	N/D	0.0160 ± 0.0007 ^b^	N/D	N/D	0.0405 ± 0.0004 ^a^	N/D
Hesperidin	0.0710 ± 0.0010 ^c^	0.1025 ± 0.0010 ^b^	0.1100 ± 0.0013 ^a^	0.0200 ± 0.0002 ^f^	0.0580 ± 0.0005 ^d^	0.1025 ± 0.0007 ^b^	0.0350 ± 0.0002 ^e^
Trans-cinnamic acid	N/D	N/D	0.0700 ± 0.0005 ^c^	N/D	0.5090 ± 0.0014 ^b^	N/D	1.4905 ± 0.0007 ^a^
Naringenin	0.0070 ± 0.0002 ^c^	0.0035 ± 0.0002 ^f^	0.0045 ± 0.0003 ^e^	0.0060 ± 0.0003 ^d^	0.0180 ± 0.0004 ^b^	0.0210 ± 0.0002 ^a^	0.0040 ± 0.0001 ^ef^
Kaempferol	0.0015 ± 0.0001 ^c^	0.0130 ± 0.0006 ^a^	0.0010 ± 0.0002 ^c^	0.0015 ± 0.0002 ^c^	0.0080 ± 0.0002 ^b^	0.0010 ± 0.0003 ^c^	0.0015 ± 0.0002 ^c^

^a–g^ Means in a row with the same lowercase letter do not differ significantly (*p* ≥0.05). N/D—not detected.

**Table 4 foods-12-02616-t004:** Quantity of phenolic compounds in parent, control, and advanced potato mutant lines (M_1_V_8_) from the fourth hybrid combination.

Concentration, (µg/mL)					
Phenolic Compound	PC 707	PC 428	K-IV-3	M-IV-14	M-IV-17
Ferulic acid	0.0230 ± 0.0004 ^c^	0.1145 ± 0.0004 ^a^	0.0290 ± 0.0007 ^b^	0.0095 ± 0.0004 ^d^	0.0220 ± 0.0003 ^c^
Rutin	0.0110 ± 0.0001 ^a^	N/D	N/D	0.0010 ± 0.0003 ^c^	0.0040 ± 0.0005 ^b^
Naringin	N/D	N/D	N/D	N/D	N/D
Apigenin	0.0440 ± 0.0002 ^c^	0.0430 ± 0.0004 ^c^	0.0595 ± 0.0004 ^a^	0.0455 ± 0.0007 ^b^	0.0410 ± 0.0004 ^d^
Hesperetin	N/D	N/D	N/D	N/D	N/D
Chlorogenic acid	3.3095 ± 0.0012 ^b^	12.5750 ± 0.0028 ^a^	2.1180 ± 0.0008 ^c^	0.7860 ± 0.0008 ^d^	2.0435 ± 0.0006 ^c^
Quercetin	N/D	N/D	0.0470 ± 0.0008 ^a^	0.0305 ± 0.0004 ^b^	0.0200 ± 0.0002 ^c^
Narirutin	N/D	N/D	N/D	N/D	N/D
Hesperidin	0.0120 ± 0.0003 ^b^	0.0275 ± 0.0004 ^a^	0.0025 ± 0.0003 ^c^	N/D	N/D
Trans-cinnamic acid	0.5425 ± 0.0009 ^b^	5.1700 ± 0.0020 ^a^	N/D	0.1185 ± 0.0006 ^c^	N/D
Naringenin	0.0035 ± 0.0001 ^d^	0.0075 ± 0.0003 ^a^	0.0065 ± 0.0003 ^b^	0.0030 ± 0.0002 ^d^	0.0050 ± 0.0003 ^c^
Kaempferol	0.0015 ± 0.0001 ^b^	0.0015 ± 0.0001 ^b^	0.0010 ± 0.0001 ^c^	0.0025 ± 0.0002 ^a^	0.0015 ± 0.0002 ^b^

^a–d^ Means in a row with the same lowercase letter do not differ significantly (*p* ≥ 0.05). N/D—not detected.

**Table 5 foods-12-02616-t005:** Quantity of phenolic compounds in the parent and the advanced potato mutant line (M_1_V_8_) from the seventh hybrid combination.

Concentration (µg/mL)				
Phenolic Compound	PC 538	PC 757	K-VII-4	M-VII-7
Ferulic acid	N/D	0.0480 ± 0.0006 ^b^	0.0310 ± 0.0002 ^c^	0.0585 ± 0.0001 ^a^
Rutin	N/D	N/D	0.0060 ± 0.0002 ^a^	0.0035 ± 0.0003 ^b^
Naringin	N/D	N/D	N/D	N/D
Apigenin	0.0410 ± 0.0007 ^b^	0.0440 ± 0.0004 ^a^	0.0405 ± 0.0003 ^b^	0.0385 ± 0.0005 ^c^
Hesperetin	N/D	N/D	N/D	N/D
Chlorogenic acid	2.4010 ± 0.0011 ^b^	1.8965 ± 0.0005 ^d^	2.7125 ± 0.0012 ^a^	1.9860 ± 0.0009 ^c^
Quercetin	N/D	N/D	N/D	N/D
Narirutin	N/D	N/D	N/D	N/D
Hesperidin	N/D	0.0635 ± 0.0010 ^a^	0.0045 ± 0.0001 ^c^	0.0175 ± 0.0003 ^b^
Trans-cinnamic acid	N/D	N/D	N/D	0.0140 ± 0.0004
Naringenin	0.0070 ± 0.0004 ^a^	0.0065 ± 0.0003 ^a^	N/D	0.0035 ± 0.0002 ^b^
Kaempferol	0.0015 ± 0.0002 ^b^	0.0010 ± 0.0001 ^c^	0.0030 ± 0.0001 ^a^	0.0015 ± 0.0001 ^b^

^a–d^ Means in a row with the same lowercase letter do not differ significantly (*p* ≥ 0.05). N/D—not detected.

**Table 6 foods-12-02616-t006:** Oxygen radical absorbance capacity assay (ORAC) of potato samples from four hybrid combinations.

Hybrid Combination	I		III		IV		VII	
	Sample	mM TE/g	Sample	mM TE/g	Sample	mM TE/g	Sample	mM TE/g
Parent Plant/s	PC 428	1.45 ± 0.21 ^c^	PC 490	2.09 ± 0.47 ^cd^	PC 707	1.84 ± 0.15 ^bc^	PC 538	1.96 ± 0.45 ^b^
	PC 490	2.09 ± 0.47 ^b^			PC 428	1.45 ± 0.21 ^c^	PC 757	2.42 ± 0.18 ^ab^
Control			K-III-2	2.07 ± 0.33 ^cd^	K-IV-3	2.16 ± 0.35 ^b^	K-VII-4	2.79 ± 0.75 ^ab^
Mutants	M-I-8	3.32 ± 0.13 ^a^	M-III-8	2.73 ± 0.50 ^bc^	M-IV-14	2.02 ± 0.22 ^b^	M-VII-7	3.08 ± 0.50 ^a^
	M-I-17	2.61 ± 0.40 ^b^	M-III-9	3.81 ± 0.30 ^a^	M-IV-17	3.03 ± 0.41 ^a^		
			M-III-30	1.55 ± 0.38 ^d^				
			M-III-48	3.84 ± 0.31 ^a^				
			M-III-50	3.41 ± 0.38 ^ab^				

^a–d^ Means in a column with different superscripts differ significantly (*p* < 0.05). TE: Trolox equivalent.

## Data Availability

The data used to support the findings of this study can be made available by the corresponding author upon request.

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
