# Peer review of "Green Assessment of Phenolic Acid Composition and Antioxidant Capacity of Advanced Potato Mutant Lines through UPLC-qTOF-MS/MS Quantification"

_foods, 2023, doi:10.3390/foods12132616_

Round 1

Reviewer 1 Report

In this study by Gomez -Urios et al, the authors characterized various mutant potato lines from the phenolic and antioxidant activity point of view through the induction of mutagenesis by EMS. The study seems interesting but some aspects need to be clarified (in particular their exploitation for commercial purposes).

Major:

The authors should indicate the possibility of commercial and food exploitation of these mutant lines and any limitations associated with their formation as obtained by mutagenesis. Is it permissible from a legislative point of view to induce mutagenesis in the described method and subsequently commercialize such products thus obtained?

One of the major doubts is inherent in the reproducibility of EMS-induced mutations since such mutations may be nonselective and affect different parts of DNA in accidental order generating random samples from each other. The authors should explain or deny (with justification) these possibilities. From how many replicates were the samples subsequently analyzed obtained?

Include the standard deviation or standard error in Tables 2, 3, 4 and 5.

Include an appropriate statistical analysis to compare the variation of individual phenolic compounds between samples and modify the text in accordance with the results obtained from the statistical analysis. In fact, the explanation of the results must be corroborated by the statistical analysis. For example, the sentence "The concentrations of kaempferol and naringenin in the mutants did not differ much from those..." is statistically meaningless: it either differs statistically or it does not. Re-check all text after performing statistical analysis

Minor:

I do not recommend referring to phenolic compounds as metabolites as it is too general and, in light of the discipline of metabolomics, may give rise to misunderstandings

It is unclear what it means to discuss the extraction process (lines 76-82) for a product that is consumed as food and that does not undergo bioactive recovery

Lines 83-91. The meaning is rather obscure. Please rewrite and explain more clearly.

It is not clear on which part of the potato the determinations were made. In section 2.4 it is clear that the extraction was performed on the whole potato while in section 2.5 the SPE is performed on the peel extract.

A figure schematizing the process of obtaining the various potato hybrids followed by the extraction process of the analyses performed is recommended in order to better understand the experimental scheme of the study

In many cases the phenolic acid content or the parent line appears higher than in the mutant lines. In their conclusions, the authors should stress more on the most promising lines and the possibility and feasibility of their commercial exploitation

Some English errors are present in the text.

Author Response

Author’s reply:

Dear Reviewer,

Thank you very much for reading our manuscript entitled “Green assessment of phenolic acids composition and antioxidant capacity of advanced potato mutant lines through UPLC-qTOF-MS/MS quantification” (Foods-2464734). The updated manuscript has been amended in accordance with the reviewer's suggestions, and the modifications have been highlighted in yellow in the text. We believe the comments were highly valuable and made the manuscript clearer and more understandable. Due to the reviewer’s comments, we also made some edits and alterations to make the manuscript more reasonable, please note that the modifications are also highlighted in yellow.

Please find a point-by-point response to each of the comments given by the reviewers.

We look forward to hearing from you at your earliest convenience.

Greetings

Reviewer 2 Report

The manuscript titled   Green assessment of phenolic acids composition and antioxi-2 dant capacity of advanced potato mutant lines through UPLC-3 qTOF-MS/MS quantificationis a continuation of earlier work by these authors and is interesting but has several shortcomings that need to be clarified or explain.

Section “Standard preparation”, is it mistake or really is possible dissolve some aglicone in 75%H2O

Line 149 , 150  it should be µg/ml (and check carefully  rest of manuscript )

Line 164 What exactly  is  5ml of sample ? More information is required. And what is control samples ?

In SPE extraction firs was used water or methanol for column conditioning?

Line 177-178 “ A. 0-5 min. isocratic 177 70% solvent A: water/formic acid 99.9/0.1 (v/v), 30% solvent B: methanol/formic acid, 178 99.9/0.1 (v/v).” it should be on the begining of description separation condition

Line 189  for quantification MS spectra were used, or UV ?

Line 312  M-IV-7?

Supplementary :

Table of validation parameters of the UHPLC-MS methodology for the determination of particular polyphenols are required.

Author Response

(The authors gave the same response as above.)

Round 2

Reviewer 1 Report

The authors have improved the overall quality of the manuscript and now it possesses sufficient quality to be published

n/a

Reviewer 2 Report

The authors revised the manuscript according to my suggestions, so I consider to  accept in this form.